# Predictive Value of Repeated Jump Testing on Nomination Status in Professional and under 19 Soccer Players

**DOI:** 10.3390/ijerph192013077

**Published:** 2022-10-11

**Authors:** Zacharias Papadakis, Vassilios Panoutsakopoulos, Iraklis A. Kollias

**Affiliations:** 1Human Performance Laboratory, Department of Health Promotion and Clinical Practice, College of Health and Wellness, Barry University, Miami Shores, FL 33161, USA; 2Biomechanics Laboratory, School of Physical Education and Sports Sciences at Thessaloniki, Aristotle University of Thessaloniki, 54124 Thessaloniki, Greece

**Keywords:** vertical jump, talent identification, biomechanical analysis, young players, anaerobic jumping power, performance, reactive strength index

## Abstract

Soccer clubs invest time and money in multidimensional identification practices, but the field implementation is still problematic. The repeated vertical jump test (RVJ), as an alternative to the monodimensional vertical jump, may offer similar prognostic value. Therefore, the prognostic validity of 15 RVJ within professional (PRO, *n* = 24) and under 19 years old (U19, *n* = 20) Greek male soccer players was examined. T-test, binomial logistic regression, and receiver operating characteristic for prognostic validity of anthropometric and performance values in predicting PRO status were applied using Jamovi version 2.3.3.0. Significant group differences presented in body height and mass, body mass index (BMI), maximum and average jump height, and relative jump power. The predicting model was significant (*x^2^*(2) = 17.12, *p* < 0.001). Height and BMI were positive predictors of the PRO status (*b* = 21.66, *p* = 0.008 and *b* = 0.94, *p* = 0.014, respectively). The model was 73% accurate, 75% specific, and 71% sensitive, with excellent area under the curve. The RVJ test demonstrated outstanding discriminating prognostic validity. Until the applicability of the multidimensional models in predicting future player status is further established, field practitioners may use the simplistic RVJ testing to predict future status among male soccer players.

## 1. Introduction

The physiological demands of soccer, as revealed by the related scientific research and notational analysis, involve high-intensity actions with short or long recovery periods as an integral part to soccer performance [1,2,3,4,5]. It is postulated that elite soccer of 2030 will involve 40% more dense periods of high-intensity actions compared to 2012/2013 season [6]. Strength and conditioning programs, to fulfill the demands of the training principle of specificity [7], are designed to provoke adaptations to soccer players based on the physical performance characteristics of the game [8]. Since the game of soccer is evolving towards higher intensity actions [6], players will have to execute dense high-intensity actions that will subsequently increase their chances to win more games [9,10].

On the other hand, soccer clubs are businesses that want to capitalize on their assets as soon as they can and prolong their players’ career to maximize the potential revenue [4,11,12,13]. Therefore, soccer clubs have invested in early talent identification programs, as early as 11 to 16 years old, to develop these youth players into elite-professional ones [4,10,13,14,15]. Williams and Reilly [16] have developed the “Potential Predictors of Talent in Soccer”, which incorporates 25 potential predictors in relation to physical, physiological, psychological, and sociological components [16]. When it comes to physiological (i.e., speed, strength, aerobic and anerobic capacities) and physical components (i.e., stature, body mass, and body composition), such programs are based on selective criteria and testing procedures that assess players’ speed, strength, aerobic and anaerobic capacities, body height and mass, and body composition. These criteria are highly influenced by the maturation levels on anthropometric, aerobic, and anaerobic capacities of the individual players [17,18]. Youth players that have been selected for professional clubs tend to have higher body mass index (BMI) values, and are taller and faster than matched ones [19,20,21,22,23,24]. In general, professional players have higher maximum oxygen consumption, faster sprint times which are executed at higher intensities, greater repeated sprint ability, larger strength back squat values, higher jumping abilities, and lower body fat percentage compared to non-professional players [25,26,27,28,29,30,31,32,33,34,35,36,37].

While these findings clearly demonstrate the multifactorial nature of distinguishing youth players that may involve to elite professional status, this is the caveat of the whole multidisciplinary approach when the predictive value of “Potential Predictors of Talent in Soccer” is applied in practice [13]. Currently, there is no scientific or practical integrated evidence of all of the Williams and Reilly suggested predictors [16,38,39]. This lack of evidence in the Williams and Reilly proof of concept paradigm [16,38,39] is that such multidimensional models are not feasible to be implemented by the field practitioners and other stakeholders due to lack of knowledge on how to evaluate and interpret the results, established scientific validity, ability to test large number of players in a short time, and available resources [38,40,41,42]. Moreover, early identification based on the multidisciplinary concept is not free of limitations. First, extensive data collection is needed, but the most prominent limitation stems from the assumption that early indicators are indeed valid predictors of future talent development that will lead to a professional status [43,44]. On top of that, the inclusion of several variables that may follow a liner or a non-linear pattern into an extensive model creates statistical interpretation problems related to multicollinearity (i.e., highly interrelated predictors) and related variance among the included variables [45,46,47].

Therefore, as of today, there are no reliable tests of future success in soccer, with research focusing on identifying the youth performance variables that may contribute to selection in professional teams [30,39,48]. In addition, there are no deselection protocols of players that are losing potential for future elite performance [39,49]. Moreover, even though the current multidimensional approach has been proposed [16,19,20,39], the majority of the clubs establish their talent identification and development programs on scout and coaches’ subjective data [13]. It is evident that there is a gap of understanding on what research is proposing and what is actually applied. This gap may be attributed to the lack of education between (a) the involved stakeholders [50] and (b) the scientific evidence for a multidimensional approach over the current utilized performance tests [13,39] and their prognostic relevance [31,39,51].

Multidimensional models present field-based implementation difficulties [15,38]. Given the current uncertainty on which method is best to predict professional success, practitioners need a less extensive, single field-based/monodisciplinary powerful single predictor/method that will classify youth’s soccer potential to a professional status [38]. Finding the most powerful single predictor will contribute to identifying relevant characteristics within a group (e.g., professionals vs. non-professionals) and to selecting the most appropriate training stimuli for future performance potential [19,52,53,54]. Assuming that the proposed paradigm of Williams and Reilly has a hierarchical order [16], this raises the question of which disciplines and related variables may be worthy of inclusion in a monodimensional modeling that will provide power discriminating values for future performance levels [15].

One of the monodimensional predictors that is extensively used to access soccer related lower limb power is the vertical jump (VJ), with its variety of different types of jumps [5,55,56]. Vertical jump assesses the soccer functional performance and at the same time is used for fitness and talent selection [57,58,59]. Vertical jump parameters, such as jump height assessed by the countermovement jump (CMJ), have been able to discriminate soccer performance levels between professional and non-professional players in a wide age range [4,19,20,60,61,62]. Essential VJ parameters are also the power output, since it is suggested to be the parameter defining VJ performance [63], as well as the reactive strength index (RSI), which is related to strength and conditioning measures and spot performance [64].

However, VJ and its variety of different types of jumps have been critiqued regarding their validity [65,66,67]. Thus, it is still unclear which jump (or type of jump(s)) has the best prognostic or diagnostic value to the involved stakeholders [55,56]. As is evident from past research, one trial of the repeated vertical jump (RVJ), specifically the 15 continuous maximal CMJ test on individuals (age 18–30 years) from different multiple and/or individual sports, is adequate to obtain valid data of any measured parameters [68,69]. In addition, anthropometric characteristics may contribute to players’ future success [5,70], with central defenders and goalkeepers tending to be the tallest and heaviest players [24,71]. On top of that, the heavier and tallest players present higher scores in VJ performance [71].

Therefore, as the multidimensional approach in predicting future soccer success is still unsubstantiated, this observational study aimed to examine whether the monodimensional RVJ test and its respective performance outcomes—measures can be accurate with high specificity and sensitivity to predict future success (e.g., professional status). It was hypothesized that a model referencing the under 19 years old soccer players (U19) category will not be able to correctly categorize the status between the U19 and professional (PRO) soccer players using anthropometric- and performance-related variables as obtained from the RVJ test.

## 2. Materials and Methods

### 2.1. Participants

A non-randomized and non-interventional design was employed to answer our research question. Twenty-four adult male professional soccer players (24.6 ± 4.7 years, 1.80 ± 0.07 m, 75.4 ± 6.7 kg, BMI: 23.3 ± 1.1 kg/m^2^, 17.9 ± 4.7 years of training experience), with participation in an international top-level competition in Greek League, comprised the PRO group. Twenty male soccer players aged less than 19 years old (17.8 ± 4.7 years, 1.70 ± 0.05 m, 67.9 ± 5.1 kg, BMI: 22.3 ± 1.3 kg/m^2^, 7.9 ± 2.7 years of training experience), playing in a semi-professional development league, served as the U19 group. The participants’ training experience was self-reported. To participate in the study, players were obliged to systematically participate in their training program and not be subjected to severe injury that compelled them to be absent from their training and competition schedule for a period of six months prior to the measurements. The participants were tested after the completion of the preseason preparatory period.

All participants provided signed informed consent. In the case of a non-adult U19 participant, a parental consent was acquired. The study was conducted following the guidelines of the Declaration of Helsinki and of the Institution’s Research Committee Ethics Code and approved by the Institutional Review Board of the Aristotle University of Thessaloniki (#7233/09.11.97).

### 2.2. Experimental Procedure and Instrumentation

To acquire the anthropometric characteristics of the participants, a Seca 220 telescopic measuring rod (Seca, Deutschland, Germany) (i.e., instrument to measure body height) and a Delmac PS400L scale (Delmac Scales PC, Athens, Greece) (i.e., instrument to measure body weight) were used. Briefly, the body height was measured after placing the participant facing away from the mounted stadiometer with the body in contact with the wall. Participants were asked to stand as tall as possible with heels together and feet evenly balanced at approximately 60 degrees. After deep inhalation, participants were asked to keep their head in a horizontal position where the hinged lever was placed into contact with the crown of the head. We recorded the height reading to the highest precision possible. Body mass was measured after asking participants to step on the scale after removing socks, jewelry, other accessories, and as much clothing as was feasible [72]. Afterwards, the warm-up section was conducted, which comprised of: (a) 8 min cycling on an 817E Monark Exercise Cycle (Exercise AB, Vansbro, Sweden), (b) 8 min of dynamic flexibility exercises with progressively increasing range of motion, and (c) six CMJ with a progressively increasing intensity from sub-maximum to maximum.

The experimental RVJ test was performed on a prototype 1.00 × 1.00 m^2^ K-Force force-plate with a ± 0.31 N measuring error (Kinvent, Montpellier, France). The participants performed one series of 15 maximum continuous vertical jumps immediately after an initial CMJ, aiming: (a) for the highest jump height (h_JUMP_) with the minimum possible ground contact time (t_C_) below 250 ms, and (b) for the fastest completion of the test. The RVJ was executed barefooted, and the arms were constantly kept akimbo. A RVJ series that consisted of five vertical jumps was provided as familiarization prior to the actual testing. In the case where t_C_ was above 250 ms, the trial was cancelled, and an additional trial was allowed to fulfill the requirements of the RVJ test.

### 2.3. Data Acquisition

Vertical ground reaction forces (vGRF) were recorded with a sampling frequency of 1 kHz. A Butterworth-type low-pass filter of second order with cut-off frequency on 80 Hz was used to reduce measurement error as defined with the methods of residuals [73]. The examination of the vGRF time series indicated the total duration of the RVJ test (*t_TOTAL_*), as well as the ground contact time (*t_C_*) and flight phases. Thus, the *t_C_* and flight time (*t_FL_*) for each separate jump were extracted. Jump height (*h_JUMP_*) was estimated using Equation (1):(1)$$h_{JUMP}=\frac{g\cdot t_{FL}^{2}}{8}$$
where *g* is the gravitational constant (9.81 m/s^2^). The mean *h_JUMP_* of the entire RVJ test (h_AVE_) was then calculated by averaging the h_JUMP_ recorded in each of the 15 vertical jumps performed in the RVJ test [69]. The reactive strength index (*RSI*), namely the ability to utilize stretching of the muscle and then change quickly from an eccentric to a concentric contraction [64], was calculated as depicted in Equation (2):(2)$$RSI=\frac{h_{JUMP}}{t_{C}}$$

The average *RSI* (*RSI_AVE_*) was the mean value of the separate *RSIs*. Finally, the average relative power output (*P_REL_*) for the whole *RVJ* was calculated according to Bosco et al. [74] (Equation (3)):(3)$$P_{REL}=\frac{g^{2}\cdot t_{FL}\cdot t_{TOTAL}}{4\cdot n\cdot \left(t_{FLsum}\cdot t_{TOTAL}\right)}$$
where *n* is the number of jumps executed within the RVJ test (=15) and *t_FLsum_* is the summation of the separate *t_FL_s* recorded.

Body mass index was calculated as in Equation (4):(4)$$BMI=\frac{bm}{bh^{2}}$$
where *bm* is the participants’ body mass in kilograms and *bh* is their body height in meters.

### 2.4. Statistical Analysis

Data were presented as mean ± standard deviation. Normality of distribution and the equality of variance were assessed using the Kolmogorov–Smirnov test (*p* > 0.05) and the Levene’s test (*p* > 0.05), respectively. A series of independent samples T-tests on anthropometric (i.e., age, height, body mass, BMI) and RVJ (i.e., hJUMP, hAVE, RSI, RSIAVE, PREL) variables of interest were performed to examine differences between the two groups with Ho: μPRO = μU19 and Ha: μPRO > μU19 (*p* > 0.05) with effect sizes (ES) of above 0.8, between 0.8 and 0.5, between 0.5 and 0.2, and lower than 0.2 were considered as large, moderate, small, and trivial, respectively [75]. Significant different values as obtained from the *t*-test were entered to construct the model(s) using binary logistic regression (BLR). Each model used BLR from R to identify the classifying variables [76,77] and receiver operating characteristic (ROC) from the R package pROC to evaluate the discrimination of the previous BLR model(s) [78,79] as previously described with proper cutoff calibration, and only the models with the highest possible number of significant predictors, with the highest accuracy and the lowest deviance and Akaike information criterion (AIC), will be presented. Variance Inflation Factor (VIF) and Tolerance were used to assess multicollinearity among the selected predictors. Omnibus Likelihood Ratio Tests for each predictor variable was used for classification purposes. Model coefficients indicated the improvement from the intercept model with likelihood ratio and odds ratio tests from group = PRO vs. group = U19, at *p* < 0.05 [80]. The likelihood of from BLR was used to develop the ROC (cut-off point value was set at 0.5) and the area under the curve (AUC), as a measurement of the quality of the classification [81,82]. An AUC of 0.5 suggests no discrimination, 0.7 to 0.8 is considered acceptable, 0.8 to 0.9 is considered excellent, and more than 0.9 is considered outstanding [83]. Predictive measures were presented as a proportion of correctly classified U19 (specificity), correctly classified PRO (sensitivity), and correct percentage of all predictive classifications (accuracy) [78]. All statistical analyses were performed using the R-based Jamovi software version 2.3.3.0 [81,84,85].

## 3. Results

Descriptive statistics are presented in Table 1. Independent samples T-tests showed that PRO and U19 soccer players were significantly different in height, mass, body mass index, maximum jump height, average jump height, and relative jump power (Table 2), with higher values recorded for PRO (i.e., H_a_ μ PRO > μ U19 at *p* < 0.05). Using the significant variables obtained from the *t*-test analysis, separate BLR models were constructed after evaluating for the collinearity assumption. The BLR model with body height and BMI as predictors was able to predict PRO from U19. The overall model was significant (*x^2^*(2) = 17.12, *p* < 0.001; see Table 3). Height was a positive and significant *(b* = 21.66, SE = 8.20, *p* = 0.008) predictor of PRO status, with OR indicting that for every one unit increase on height, the odds of receiving PRO status changed by a factor of 2,556,543,943.10, meaning that the odds are increasing (see Table 3 and Figure 1). BMI was a positive and significant *(b* = 0.94, SE = 0.38, *p* = 0.014) predictor of PRO status, with OR indicting that for every one unit increase on BMI the odds of receiving PRO status change by a factor of 2.57, meaning that the odds are increasing (see Table 3 and Figure 2). Across both outcome categories, 73% of the cases were accurately classified, with specificity somewhat equal to sensitivity. U19 soccer players were 75% correctly predicted compared to 70.83% of the correctly predicted PRO players (see Table 4), with an AUC of 0.82, which is considered excellent (see Table 4 and Figure 3).

## 4. Discussion

The aim of this study was to evaluate whether a monodimensional talent identification and future prediction model approach using the 15 RVJ test could accurately discriminate and predict the status between PRO and U19 male soccer players. After running several BLR models on anthropometric and RVJ lower limbs strength and power variables, grounded by the AUC results, an excellent discriminatory prognostic validity of PRO status based only on body height and BMI (i.e., AUC of 0.50 = no discrimination; 0.70 = acceptable discrimination; 0.80 = excellent discrimination; and 1.00 = perfect discrimination) was indicated [83,86]. None of the examined models that included the RVJ related variables (i.e., maximum and average jump height, reactive strength index, relative power) or in combination with the anthropometric ones (i.e., body height and mass, body mass index) resulted in significant predictors. The next model, including only one predictor, was the body mass with accuracy 73%, specificity 75%, sensitivity 71%, 83% AUC, similar with our two predictors model of height and BMI, but with higher deviance (43.56) compared to our model (43.52), pointing to a reduced model fit measure. The second less fit model included the maximum jump height as a single significant predictor with accuracy 64%, specificity 67%, sensitivity 60%, 70% AUC, and deviance of 54.05, highlighting a lesser fit model. The comparison of these two single predictor models with the double predictor model that we presented supports the advantage of the complexity and multidimensionality over the simplicity and monodimensionality [16,19,20,21,24,28,39,60]. However, as previously described, the multidimensional superiority regarding talent identification is hindered by its field applicability over the simplicity of the monodimensional concept within the talent research, something that researchers need to acknowledge [16,38,39,40]. Similar to our findings, Sieghartsleitner et al. [38], using BLR modeling on general or single motor components, failed to differentiate between professional and non-professional players. The authors, like in our case, pointed out that the more extensive modeling, within the similar group of participants, provided higher explained variance and higher prognostic validity [38].

Moreover, the discriminatory excellent AUC (0.83) ability of our two predictors model within the PRO and U19 is partially in line with earlier research based on German football talent identification and development program [48,87]. Honer et al. [48] verified the prognostic ability of both physical performance and anthropometric variables over a 9-year period in a sample of 14,178 soccer players. In their study, anthropometrics such as weight and height were significant predictors to players’ future success [48], in contrast to our study, which failed to provide evidence of the physical performance predictors. Our reported AUC is much higher compared to the reported AUC that ranged from 0.65 to 0.79 when general motor performance (i.e., 40 m sprint, agility, CMJ, Yo-Yo intermittent recovery test) and domain-specific motor performance parameters were used as predictors of future U17 players’ status [38]. The difference in the age groups and the associated maturity levels between the forementioned study and the present study may be the reason we reported higher AUC, since both age and maturity have a significant impact on prediction models [15,29,39,88,89].

Regarding the value of the predictors, body height had by far the highest impact on predicting the PRO status with an extraordinary odds ratio (*OR* = 2.56 × 10^9^ [266.23, 2.5 × 10^16^]), indicating that for every one unit increase on the height predictor the odds of PRO change by a factor of 2.56 × 10^9^. A similar trend, with lower magnitude, was presented for the BMI (*OR* = 2.57 [1.21, 5.44]), highlighting that, as BMI increases by one unit, the odds of PRO change positively (increase) by a factor of 2.57. Both of these predictors show the significance of anthropometrics and the dependence on the biological maturation, even though there is great heterogeneity in body size among elite soccer teams [4,28,29,40,90,91]. This is partly due to the playing position, as goalkeepers, central defenders, and central attackers tend to be the tallest and the heaviest players of a soccer team [5,71,92,93,94,95]. In addition, Gil et al. [96] presented that besides the speed, aerobic capacity, and chronological age, height and size are also associated with predicting whether or not young soccer players (14–17 years old) will be successful or not as soccer players [96]. Gil et al. [95], in a study aiming to establish an anthropometric and physiologic profile of soccer players based on player position, reported that regarding BMI scores, goalkeepers had the highest ones, similar to overweight sedentary individuals, while defenders were discriminated by lower leg power [95]. Similar to Gil et al. [96], the Lago-Penas et al. group examined the relationship between playing position, anthropometrics, and physiological profile, predicting competition success in young soccer players (range 12–19 years), and reported that successful players were taller, heavier, leaner, and more muscular compared to non-successful counterparts [71,97]. Elite and non-elite U19 Portuguese male soccer players’ comparisons revealed that elite players were taller and heavier than non-elite players, with body size differences to discriminate position and competition level [62]. Regarding the anthropometric parameters, selected elite Italian soccer players were taller and had higher BMI compared to the non-selected and non-elite counterparts [5]. Saward et al. [41] examined the association of youth elite soccer players (*n* = 2875) from the English talent development system who achieved PRO status whether or not they differed in stature, body, mass, and physical performance compared to non-professional players. They found no differences in body mass and stature, but they reported that future professionals, as assessed with the CMJ test, were jumping higher than the non-professionals.

In our study, none of the RVJ parameters were significant predictors of PRO status. In agreement with our study, Castagna and Castellini [98] examined the predictive validity of VJ in 56 Italian national team-selected male soccer players and concluded that VJ could not discriminate between competitive performance levels. Moreover, they reported that the ROC curve analysis showed no threshold values [98], while our study’s ROC curve presented thresholds and had excellent AUC pointing model’s ability to accurately detect as many true positives as possible, minimizing the false positives between U19′s future PRO status [83,99]. Similarly, when 34 male professional soccer players from the Spanish League were divided into elite and non-elite levels of competition and senior (24 years) and junior (18 years) based on age, authors reported no discriminant power for level of competition based on VJ performance [100]. The biological maturation and the absence of heterogeneity over the physical performance variables between the PRO and U19 may be the reason why this study failed to provide prognostic future status based on RVJ performance parameters [101,102]. Furthermore, the time period where the testing was performed [103,104], or even the individual match playing time [105], may have been a determinant factor for not discriminating VJ performance between PRO and U19. Graig and Swinton [106] followed 512 players U10 to U17 for 10 years to predict success based on four different types of VJ in elite Scottish soccer players. Even though they found that, on average, successful players were taller and performed better in the jump tests, these differences were not able to predict success. Authors attributed this phenomenon to the different VJ applied protocols [106], something that is not applicable to the present study, as our experimental protocol has been shown to have high reliability [68,69].

The predictive ability of the RVJ in our study is in contrast with what Arnason et al. [107] and Wisloeff et al. [108] presented using elite male Scandinavian players. Both groups reported that VJ performance, as assessed with the squat jump CMJ, was able to discriminate differences between groups in favor of the higher ranked team players due to the homogenous sample of players. Similar to the previous studies, CMJ testing was recommended to better distinguish elite and sub-elite U16 male soccer players [109], while squat VJ was the strongest predictor of playing status in Australian soccer [93]. Rebelo et al. [62] evaluated whether anthropometric characteristics, fitness (i.e., squat jump, CMJ, strength, Yo-Yo, 5- and 30-m sprints) and performance related variables of Portuguese U19 elite and non-elite soccer players can have a discriminative significance. They noted position-specific differences in size (body height and body mass), with the discriminative effect to be primarily in elite goalkeepers and central defenders [62]. This investigation reported differences in size as well, but due to our research question and characteristics of our sample we did not perform a position-specific analysis.

It should be noted that irrespective of the results of this study and its agreement with the literature, it would be simplistic to go against the majority of the evidence that supports that anthropometric and/or physiological parameters alone can determine success in soccer [39]. A recent study, which examined technical, tactical, physical, and psychological parameters on Finnish soccer players and their ability at the age of 15 years to predict professional status at age of 19 years, presented a multidimensional model with 86% discriminative validity [110]. In general, achieving professional status in soccer is dependent on the likelihood of having higher cardiorespiratory endurance, faster sprinting times, greater sprinting abilities, greater vertical jumping capacities, higher one repetition back squat values, higher lean mass and lower body fat percentage, being taller and heavier as youth players, and lastly being born in the first semester of the year [10,17,19,39,40,43,51,60,90,91,110].

This study is limited by the small sample size, and by its basic concept of examining a multidimensional task under a monodimensional perspective. Authors recognize that soccer requires complex skills (e.g., ball handling, tactical knowledge, psychological capacities) other than speed, strength, and power to be successful [25,26,27,28,29,30,31,32,33,34,35,36,37]. Williams and Reilly’s [16] work has advanced the area of talent identification and development towards a multidisciplinary approach that measures a number of predictors across key aspects of soccer performance [39,60] and led other researchers to further research this topic [20,50,110,111]. Moreover, the premise of using a multidimensional process to predict future player status is to minimize the risk of false selection decisions [19,38,40]. However, at the same time, this approach has been problematic in applied fields and abandoned by coaches and scouts, as there is no study that has integrated all the suggested predictors, which led to the inception of this study [6,15,38,39,40]. As this study is cross-sectional and the most common approach in the area [15], the understanding of the prospective value of our selected predictors and how these involve over time due to maturation and exposure to systematic training cannot be assessed [39,51,112]. Another limiting factor is that our results are sex-specific and cannot be discussed from a female perspective, or aid in recognizing the sex-related differences in identification and developmental process in soccer [39,57,98,113,114]. Moreover, cross-sectional studies’ predictive value and validity have been questioned and their usefulness to provide answers has been distrusted [15,39]. Differences in maturation and development, as well as subsequent physiological and anthropometric characteristics changes, may lead to false negative discrimination/classification by excluding late-developing athletes. These late-developed athletes could have had the potential to be successful as PRO [19,94,110], since development does not have linear behavior [15,16,19,115], which may require non-linear statistical analysis [115,116]. On top of that, we chose not to analyze our data based on position, as there was not a balanced design and we wanted to be able to provide a simple interpretation of the results. In addition, the highly selected individuals limit our results by comprising a homogenous sample where the selected predictors had limited discriminatory power, as this created a range restriction [39,117]. Of course, the fact that our current modeling had limited discriminatory value does not imply that this model will have the same behavior when a broader sample of players is used [39] or that our selected predictors are not important for soccer success [118]. Lastly, our statistical analysis was based on BLR modeling. Therefore, it may suffer from its related assumptions and weakness to detect intra-individual interactions between predictors, where the weakness of one predictor is compensated by the strength of another [119].

A strength of this study was the use of the BLR and ROC modeling that can provide insights into the non-linear components of soccer performance. Soccer performance is rarely linear, as it is influenced by both cognitive and motor skills that are developed through dynamic interactions with players’ environment [117]. Therefore, since there is a complex non-linear environment present that influences players’ future development and success, statistical modeling through BLR and ROC have been deemed as appropriate [120]. BLR examined the probability of PRO status based on U19 selected anthropometric and physiological predictors [77,82,86,90,110,121], while ROC was used to show the tradeoff between sensitivity and specificity and to present the accuracy of the predicting model [83,99,122,123]. Such statistical modeling provides better evaluation of the variance in the performance, balancing the advantages and disadvantages of the selected predictors [39,40,78,99,110,111,124,125].

In the future, researchers need to come to an agreement on which standardized methods/tools are the best for assessing soccer performance in the field [39]. Such a standardized method can lead to studies with higher predicting and accuracy models that are specific to certain ages, playing positions, and sexes, while at the same time allowing for longitudinal performance tracking [39,117,126]. Since maturity, chronological age, and relative age are key elements for soccer identification and development, this standardized tool needs to account for these factors [39,127]. Moving forward and based on the current technological advancement, fields like neuroscience and genetics may be able to provide additional monodisciplinary evaluation of the genetic makeup of successful soccer players [39]. For example, a study that examined the genetic makeup of Russian soccer players concluded that compared to control population, players had greater incidence of four alleles that are linked to power/strength, endurance, metabolic efficiency, and muscle hypertrophy [128]. In addition, height and skeletal size present hereditary characteristics [129], something that can be used in the talent identification and future development and success. Therefore, all the involved stakeholders in the player recruitment, identification, and development should come to an agreement and work together to identify the criteria used for the most effective field applied practice [39]. Lastly, when it comes to research practices regarding talent identification and future soccer success, a proper selection of individual performance outcomes with high predicting fidelity and correction for range restriction while accounting for baseline improvement and examining longitudinal and/or prospective basis is needed [15,39,51,90,117].

## 5. Conclusions

Based on the reported results of this study and on the vast evidence in the literature, it appears that elite professional soccer players present specific attributes and characteristics in comparison to non-elite players and control population [10]. Using a simplistic monodimensional approach such as the RVJ, field practitioners can be assured that they are using a highly accurate and specific method to predict future success in soccer. As yet, field practitioners need to recognize that there is no simple and single objective measure that can, in a holistic way, capture the complexity that surrounds the sport of soccer. Individual performance differences are dependent on soccer itself and its regulations, which make any prediction modeling extremely challenging [130].

## Figures and Tables

**Figure 1 ijerph-19-13077-f001:**
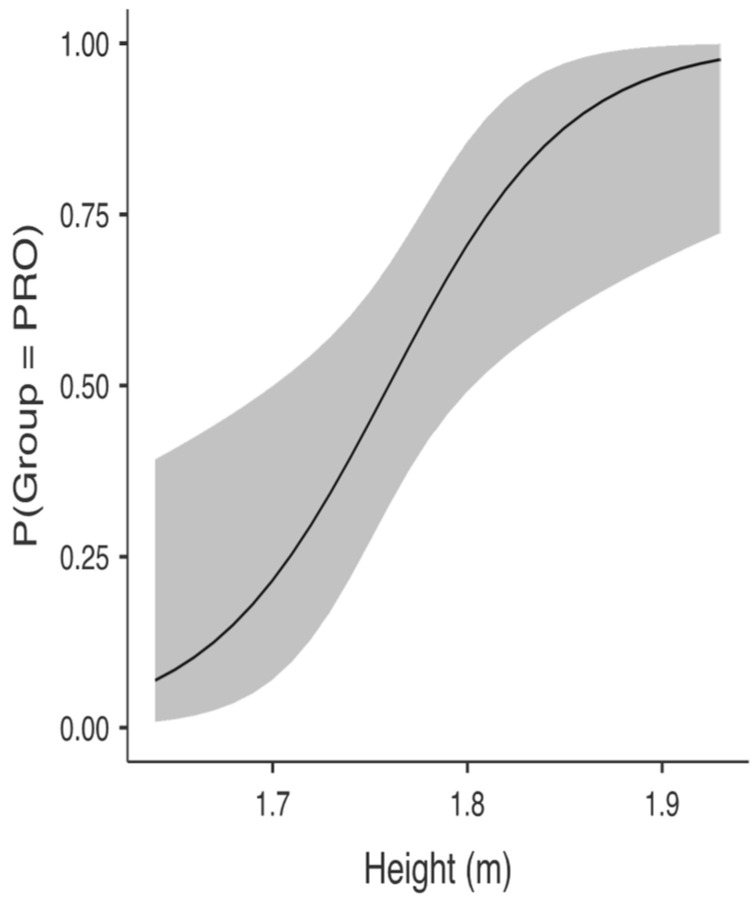
U19 body height probability to become professionals (PRO).

**Figure 2 ijerph-19-13077-f002:**
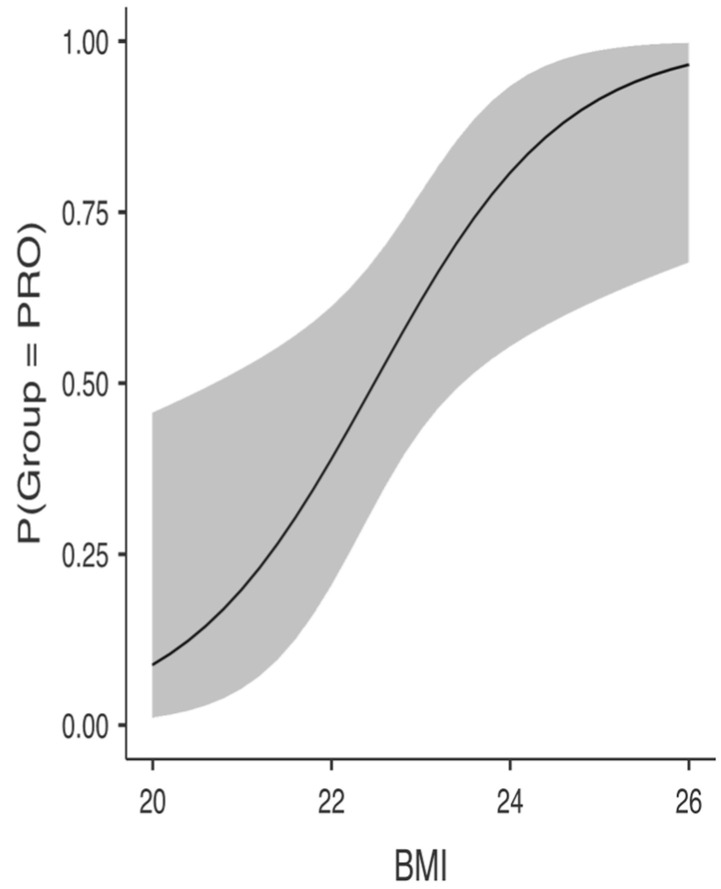
U19 body mass index (BMI) probability to become professionals (PRO).

**Figure 3 ijerph-19-13077-f003:**
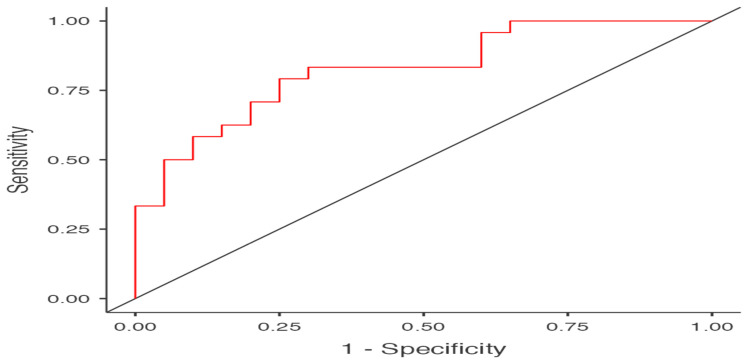
Receiver Operating Characteristic Curve.

**Table 1 ijerph-19-13077-t001:** Descriptive Statistics.

	Group	N	Mean	SD
Age (yrs)	PRO	24	24.63	4.67
	U19	20	17.77	1.51
Height (m)	PRO	24	1.80	0.06
	U19	20	1.74	0.05
Mass (kg)	PRO	24	75.40	6.69
	U19	20	67.90	5.08
BMI (kg/m^2^)	PRO	24	23.25	1.13
	U19	20	22.30	1.27
Jh_MAX_ (m)	PRO	24	0.32	0.04
	U19	20	0.28	0.05
h_AVE_ (m)	PRO	24	0.26	0.04
	U19	20	0.22	0.06
RSI_MAX_ (m/ms)	PRO	24	1.88	0.28
	U19	20	1.78	0.40
RSI_AVE_ (m/ms)	PRO	24	1.52	0.2
	U19	20	1.41	0.33
P_REL_ (W/kg)	PRO	24	11.00	0.95
	U19	20	10.12	1.38

All values are presented as mean and standard deviation (SD). BMI: body mass index; Jh_MAX_: maximum jump height; h_AVE_: average jump height; RSI_MAX_: maximum reactive strength index; RSI_AVE_: average reactive strength index; P_REL_: relative jump power.

**Table 2 ijerph-19-13077-t002:** Independent Samples T-Test and Cohen’s d Effect Size.

								95% Confidence	
								Interval	
Parameter	*t*	df	*p*	Mean	SE	Cohen’s *d*	Effect Size	Lower	Upper
				Difference	Difference				
Age (yrs)	6.29	42	<0.001	6.86	1.09	1.90	Large	1.09	2.70
Height (m)	3.07	42	0.002	0.05	0.02	0.93	Large	0.27	1.57
Mass (kg)	4.12	42	<0.001	7.50	1.82	1.25	Large	0.54	1.93
BMI (kg/m^2^)	2.64	42	0.006	0.95	0.36	0.80	Moderate	0.15	1.43
Jh_MAX_ (m)	2.64	42	0.006	0.04	0.01	0.80	Moderate	0.15	1.43
h_AVE_ (m)	2.33	42	0.012	0.04	0.02	0.71	Moderate	0.07	1.33
RSI_MAX_ (m/s)	0.98	42	0.167	0.10	0.10	0.30	Small	−0.31	0.89
RSI_AVE_ (m/s)	1.4	42	0.084	0.11	0.08	0.42	Small	−0.19	1.03
P_REL_ (W/kg)	2.49	42	0.008	0.88	0.35	0.75	Moderate	0.12	1.38

Note. H_a_ μ PRO > μ U19. BMI: body mass index; Jh_MAX_: maximum jump height; h_AVE_: average jump height; RSI_MAX_: maximum reactive strength index; RSI_AVE_: average reactive strength index; P_REL_: relative jump power.

**Table 3 ijerph-19-13077-t003:** Model fit Measures, Omnibus Likelihood Ration Tests, and Model Coefficient.

Model Fit Measures				Omnibus Likelihood Ratio Tests											
	Overall Model Test							Model Coefficients—Group						95% Confidence Interval	
Model	x^2^	f	*p*	Predictor	x^2^	df	*p*	Predictor	Estimate	SE	Z	*p*	Odds Ratio	Lower	Upper
1	17.12	2	<0.001					Intercept	−59.63	19.88	−3.00	0.003	0.00	0.00	0.00
				Height	10.37	1	0.001	Height	21.66	8.20	2.64	0.008	2.56 × 10^9^	266.23	2.5 × 10^16^
				BMI	8.21	1	0.004	BMI	0.94	0.38	2.46	0.014	2.57	1.21	5.44

Note. Estimates represent the log odds of “group = PRO” vs. “group = U19”. BMI: body mass index; PRO: Professionals; U19: Under 19 years old.

**Table 4 ijerph-19-13077-t004:** Prediction Classification and Predictive Measures.

Classification Table				Predictive Measures			
	Predicted						
Observed	U19	PRO	% Correct	Accuracy	Specificity	Sensitivity	AUC
U19	15	5	75.00	0.73	0.75	0.71	0.82
PRO	7	17	70.83				

Note. The cut-off value is set to 0.5. PRO: Professionals; U19: Under 19 years old; AUC: Area under the curve.

## Data Availability

The data that were used in the present study can be provided by the corresponding author upon reasonable request.

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
