# Peer review of "Predictive Value of Repeated Jump Testing on Nomination Status in Professional and under 19 Soccer Players"

_ijerph, 2022, doi:10.3390/ijerph192013077_

Round 1

Reviewer 1 Report

The authors conducted an interesting study that explored the importance and impact of certain physiological and anthropometric parameters in talent identification in soccer. The sample examined included professional Greek soccer athletes competing at the international level as well as semi-professional U19 players. However, based on the stated comments and observations, it is not difficult to infer that this research requires numerous modifications before further serious consideration. More specifically, a lot of gaps and omissions have been recorded in nearly all segments of the analyzed study. 

·         The statement referring to the research objective is lacking in the abstract! Please add the missing assertion!

·         The number of keywords currently available in the manuscript is truly controversial! Please remove some of them from the text!

·         Lines 29-36: The readability, clarity, and manner of the expression in this paragraph are more than questionable! Please make the required corrections!

·         Lines 45-48: The second part of this statement appears to be an independent claim. Please justify yourself! In addition, a phrase such as “tend to have a higher body mass index (BMI) values” is certainly a more suitable option than the one presently available in the text.

·         Line 59: Did you mean "and other stakeholders"? Please provide the missing term!

·         Line 80: This statement needs to be rephrased! Several mistakes in the manner of expression have been recorded here! Please check this assertion thoroughly again!

·         Line 93: The abbreviation for the vertical jump needs to be provided when this term has been mentioned for the first time in the text! Please check the previous statement again!

·         Line 101: "The best prognostic or diagnostic values" is a most likely superior choice compared to the current expressions in the manuscript. Please consider this suggestion!

·         Lines 102-104: It would be useful to specify which populations these studies were conducted on. Please provide this information, if possible!

·         What was the design of this study? Please spread the methods section, adding data pertaining to the research design!

·         Line 120: Since the term, PRO was utilized for the first time after the abstract section, the authors should explain the full meaning of this abbreviation! The same mistakes have been repeated in the following lines and with the following abbreviations: 176: AIC; 221-223: AUC.

·         How did you obtain values for variables such as body mass index and training experience of the participants? Please report!

·         Lines 133-135: Please state more specifically which anthropometric characteristics were measured with the applied instruments. Additionally, can you please describe the protocol referring to the measurement of the anthropometric parameters? This information is essential for a clearer understanding of the entire "Measurement" section!

·         It has been noted that there is a strong connection between the names of the subtitles "Data analysis" and "Statistical analysis." In order to avoid unnecessary confusion, it would be desirable to change the name of the first stated subheading ("Data analysis"). Please consider this recommendation!

·         Lines 149-161: Have you described the parameters assessed during the vertical jump test protocol in this subtitle? Please highlight this fact! Furthermore, the authors must provide definitions for all the evaluated variables/basic concepts of this research (h JUMP, RSI, and P REAL). Please try to fulfill the stated requirements!

·         Lines 190-193: Please report which of the two groups had significantly higher values of the stated anthropometric variables and jump parameters. Moreover, the hypothesis currently available below in Table 2 should be presented within this statement.

·         Lines 196-199: Did you mean "each unit"? The same error has been repeated in the following statement! Please provide the required corrections! Further, did you mean Table 3 instead of Table 2? Please check this again!

·         Indeed, numerous unnecessary parameters of the independent sample t-test are shown in Table 2. More specifically, the authors are recommended to present the following aspects: t-test, p-values, Cohen's d, and 95% Confidence Interval! In addition, the scientists can consider merging Table 1 and Table 2 with the aforementioned parameters of the independent sample t-test!

·         Lines 214-215: "Bodyweight" or similar is surely a more suitable expression compared to "mass". This mistake has been repeated several times across the entire manuscript! Please make indispensable corrections!

·         In general, Table 2 seems confusing and unclear! Please improve the entire image of this table!

·         Line 267: Is a term such as "respectively" missing in the parentheses? Please check this again!

·         Lines 274-303: The researchers most likely presented a redundant number of previous studies' findings referring to the influence of anthropometric variables! Please consider shortening this paragraph by removing some of the mentioned investigations!

·         Lines 326-327: The noted manner of expression in this statement is quite questionable. Please provide the required changes!

·         Lines 336-338: What about findings referring to the predictive values of the vertical jump test in this investigation? Please report! 

·         Line 340: "Naive" likely represents an unsuitable term! Please consider removing this expression from the text and try to use more appropriate terminology! 

·         Line 357: Do you really consider it correct to begin a statement with conjunction? Please make the required corrections!

Author Response

Dear Reviewer #1, thanks for your valuable feedback. Please see the attached the old version with Track changes and Response to Reviewers #1, #2, #3. We are attaching also a clean final version of the manuscript.

Reviewer 2 Report

The strength of the research lies in the fact, that it tries to assess the abilities and selection of football players with a motor test, which is complemented by the measurement of anthropometric data. This test is a good measure of the leg's dynamic muscle strength and its speed endurance, because of this reason the method and the research itself are valuable, but several limiting factors must also be taken into consideration.

What is the strength of the research is also, in my opinion, the limitation of it. This should be mentioned and explained further. Its strength is that it tries to make predictions in the field of football by measuring a test. This is also a limitation of the research, as football requires complex skills, and in order to achieve success and good performance, it is important to have adequate conditional skills, which are well measured by the jump test, however, ball handling techniques (possession, passing, dribbling, ball- foot contact), tactical knowledge, etc. also determine good performance, which this test cannot measure. This is in fact mentioned in the study as well, stating that the good performance of different positions (e.g., goalkeepers, midfielders, etc.) in football, or the prediction of said performance, may differ due to the specialties.

I miss the explanation and elaboration of the limitations of the research since the other areas are well developed in the article.

Author Response

Dear Reviewer #2, thanks for your valuable feedback. Please see the attached the old version with Track changes and Response to Reviewers #1, #2, #3. We are attaching also a clean final version of the manuscript.

Reviewer 3 Report

Article 

Predictive Value of Repeated Jump Testing on Nomination Status in Professional and Under 19 Soccer Players

Abstract

This section meets sufficiently well the expectations of the future reader since it explains the main features of the research that has been carried out.

However, please note the following two observations: 

1) Lines 12-14, please explain if only one data collection has been carried out, there have been several data collections, it has been carried out at various times... or how the measurement protocol has been.

2) Lines 14-15, although the template of the journal allows it, it is a spelling mistake to break the syllables of a word, please CORRECT IT HERE AND IN THE REST OF THE TEXT.

Introduction

EXCELLENT!!!... It is a section that includes current and very well focused quotes that invite the future reader to continue reading the article. It also presents this literature review from a very general exposition to focus on current information about the research problem posed in the paper: "to examine whether the monodimensional RVJ test and its respective performance outcomes - measures can be accurate with high specificity and sensitivity to predict future success (e.g., professional status)".

2. Materials and Methods 

2.1. Participants

Participants have been clearly described both anthropometrically and in terms of training history.

Ethical standards have been respected and your research has been endorsed by an external Ethics Committee but is only indicated as being from one institution, please clearly include the name of that institution.

Experimental Procedure and instrumentation

This section describes very well what the participants have done and how it has been evaluated. 

Please inform me in which units each variable was measured, e.g. weight: kg, height: cm, jumping height: cm....

Please also inform about the rest times between each of the tests performed.

2.3. Data analysis

Well explained but missing the units of measurement of: hJUMP; RSI; PREL.

2.4. Statistical analysis

Good work, good explanations.

3 Results

The results are explained in only 15 initial lines for all tables and figures. My recommendation is to be a bit longer in your explanations and to include comments before each table or figure to help the reader to find the information next to the table or figure.

4. Discussion

This is undoubtedly the best section of the whole article where they compare the results of their work with those referenced in the introduction and others that have been newly incorporated.

5. Conclusions

It is very much appreciated that it is focused from a practical point of view so that coaches and clubs can make the best future selection of players.

ANNOTATION

One of the strong limitations of this study is that BMI has been used instead of kg muscle mass and % fat mass, nowadays these anthropometric values are of enormous value and relatively easy and cheap to obtain (bioimpedance), giving much, much, much more precision to the importance of body composition than BMI. Please consider including this limitation in your article.

Author Response

Dear Reviewer #3, thanks for your valuable feedback. Please see the attached old version with Track changes and Response to Reviewers #1, #2, #3. We are attaching also a clean final version of the manuscript.

Round 2

Reviewer 1 Report

Nice job.